# The Role of Growth Factors in the Pathogenesis of Dengue: A Scoping Review

**DOI:** 10.3390/pathogens11101179

**Published:** 2022-10-13

**Authors:** Victor Edgar Fiestas Solórzano, Raquel Curtinhas de Lima, Elzinandes Leal de Azeredo

**Affiliations:** Viral Immunology Laboratory, Oswaldo Cruz Institute, Rio de Janeiro 21040-360, Brazil

**Keywords:** dengue, severe dengue, growth factor, review

## Abstract

Growth factors (GFs) have a role in tissue repair and in the modulation of the expression of inflammatory cells in damage caused by pathogens. This study aims to systematize the evidence on the role of GFs in the pathogenesis of dengue. This scoping review considered all published peer-reviewed studies in the MEDLINE and Embase databases. Ultimately, 58 studies that analyzed GFs in dengue patients, published between 1998 and 2021, were included. DENV-2 infection and secondary infection were more frequent in the patients studied. ELISA and multiplex immunoassay (Luminex) were the most used measurement techniques. Increased levels of vascular endothelial growth factor, granulocyte–macrophage colony-stimulating factor, granulocyte colony-stimulating factor, transforming growth factor beta, and hepatocyte growth factor as well as reduced levels of platelet-derived growth factor and epidermal growth factor were observed in severe dengue in most studies. Vascular endothelial growth factor and hepatocyte growth factor were identified as biomarkers of severity. In addition, there is evidence that the dengue virus can use the growth factor pathway to facilitate its entry into the cell and promote its viral replication. The use of tyrosine kinase inhibitors is an alternative treatment for dengue that is being studied.

## 1. Introduction

Dengue is a mosquito-borne viral infection caused by dengue viruses (DENV-1, DENV-2, DENV-3, and DENV-4), the incidence of which has increased dramatically worldwide in recent decades [1]. Dengue virus infection presents with a wide clinical spectrum that includes both asymptomatic and symptomatic clinical conditions. It can progress to severe life-threatening disease, mainly attributed to increased vascular permeability, leading to plasma leakage and shock [2]. The pathogenesis of dengue virus infection involves complex interactions between the virus, host genes, and host immune response. Indeed, understanding the immunopathogenic mechanisms remains a challenge despite decades of research [3]. The excessive production of inflammatory mediators, including cytokines, chemokines, and growth factors, is associated with progression to severe dengue [4].

Growth factors (GFs) are signaling proteins that stimulate a variety of cellular processes including cell growth, differentiation, proliferation, survival, inflammation, and tissue repair [5]. The binding of GFs to their specific receptors results in the activation of their intracellular protein kinase domain, which initiates a cascade of signaling events. However, these receptors have also been identified as necessary for the entry of some viruses, and GF-receptor signaling is involved in viral replication [6].

Dengue has no specific treatment, and vaccines still have inconsistent effectiveness against all four dengue serotypes [7]. Thus, drugs targeting viral targets or critical host mechanisms are required to reduce severe illness and death due to dengue [8]. Accordingly, a better understanding of the role of GFs in dengue pathogenesis is required if treatments targeting these signaling proteins are to be developed.

GFs include vascular endothelial growth factor (VEGF), granulocyte–macrophage colony-stimulating factor (GM-CSF), granulocyte colony-stimulating factor (G-CSF), transforming growth factor beta (TGF-β), platelet-derived growth factor (PDGF), epidermal growth factor (EGF), hepatocyte growth factor (HGF), and fibroblast growth factor (FGF), among others. Primary studies analyzing different GFs in patients with dengue have been published; however, these studies have not been mapped. A preliminary search of PROSPERO, MEDLINE, and Embase databases revealed no systematic or scoping reviews on this topic.

Therefore, this study aimed to systematize the evidence on the role of GFs in the pathogenesis of dengue and support the design of new studies on dengue treatment.

## 2. Results

In total, 747 records were identified in the bibliographic search. After removing duplicates and reviewing titles and abstracts, 75 articles were selected for full-text review. Of these, 58 were eligible for data extraction. The details of the literature selection process are shown in a flow diagram based on PRISMA–ScR [9] (Figure 1).

For this review, all eligible studies were included with no publication date limits: one study was published in 1998, another in 1999, and the others between 2005 and 2021. The full texts of 56 of the articles were published in English and the remaining two in Spanish.

Most studies were performed in Asian (40/58, 69%) and American (16/58, 28%) countries, mostly in India and Brazil, respectively. The study design was more frequently cross-sectional (34/58; 59%) and prospective (18/58; 31%). The participants in most of the studies were stated as adults (27/58; 47%), followed by adults and children (17/58; 29%) and children only (12/58; 21%), while two studies did not report the age group of the participants.

Most studies used the WHO dengue case classification (48/58; 83%), and more than half (31/48; 65%) used the 1997 WHO classification that defines symptomatic dengue virus infections as dengue fever, dengue hemorrhagic fever, or dengue shock syndrome [10], even after the publication of the WHO 2009 classification that differentiates patients with dengue without warning signs; dengue with warning signs (abdominal pain, persistent vomiting, fluid accumulation, mucosal bleeding, lethargy, liver enlargement, increasing hematocrit with decreasing platelets); and severe dengue (dengue with severe plasma leakage, severe bleeding, or organ failure) [11].. The rest of the ten studies used a classification according to the objectives of the study

The serotype and type of infection were identified in 52% (30/58) and 48% (28/58) of studies, respectively. DENV-2 (22/30; 73%) and secondary infections (21/28; 75%) were more frequent in the studies that performed this analysis. DENV-1 (7/30; 23%), DENV-3 (6/30; 20%), and DENV-4 (4/30; 13%) were also identified.

GF analysis was the primary objective in 83% of the studies (48/58) and was performed with a single sample in most studies (38/58; 66%) using plasma (30/58; 52%) or serum (27/58; 47%). One study did not report the type of sample used. The samples were analyzed using ELISA (28/58; 48%) or multiplex immunoassay (Luminex) (27/58; 47%). Most studies included a healthy group as control (46/58; 79%); four studies used other febrile illnesses as controls, and eight studies did not include any control.

The results of these studies are presented below (see also Table 1). As mentioned above, the studies included in the review used different patient classifications. Thus, to facilitate the analysis, severe dengue was defined as patients classified as having dengue hemorrhagic fever (DHF) or dengue shock syndrome (DSS) according to the 1997 WHO classification [10], severe dengue according to the 2009 WHO classification [11], or severe cases established by the study itself.

### 2.1. Vascular Endothelial Growth Factor (VEGF)

VEGF was the most widely studied growth factor. A total of 33 studies analyzed VEGF, and 22 found increased levels of VEGF in patients with dengue, most of them severe dengue, compared with nonsevere dengue and/or healthy controls [12,13,14,15,16,17,18,19,20,21,22,23,24,25,26,27,28,29,30,31,32,33], including 5 studies in children only [16,17,21,26,27]. This trend of increased levels was maintained regardless of the disease classification, age group, or sample type. The median concentration range was between 54.6 and 1885 pg/mL by ELISA [14,15,17,22,24,25,28,30] and was 67.6–270.5 pg/mL using the Luminex method [19,21,23,26].

VEGF was also identified as a biomarker of severity with area under curve (AUC) values between 0.84 and 0.92 based on receiver operating characteristic (ROC) curve analysis on the second and third days of illness [20,25].

The increase in VEGF levels was evident mainly in the febrile and critical phases [12,13,14,16,18,19,20,24,26,27,33], reaching a peak at the critical phase [16,24,26,27]. Higher levels of VEGF were found in secondary infections [25,26], and in patients with comorbidities [23]. In addition, positive associations with hematocrit values [28], transaminase levels [25,28], and D-dimer levels [30], as well as inverses association with platelet count were reported [25,28].

Seven studies reported lower levels of VEGF in dengue patients than in healthy controls [34,35,36,37,38,39,40], including dengue with warning signs [35,38] and severe dengue [34,36,39,40]. No difference in results was found between serum and plasma samples [39]. Three studies found no difference between severe and nonsevere dengue [41,42,43], and one study found no difference between dengue patients and healthy controls [44].

Some studies analyzed soluble VEGF receptors (sVEGFR), sVEGFR1, and sVEGFR2. Increased levels of sVEGFR1 [16,27,40,44] and decreased levels of sVEGFR2 [16,24,27,32,40,44] were found in severe dengue. The highest levels of sVEGFR1 and the lowest levels of sVEGFR2 were found in the critical phase and were restored during convalescence [27]. Decreased levels of sVEGFR2 are associated with higher plasma viral load and increased pleural fluid at the critical phase [27]. One study found that dengue patients without plasma leakage with and without hemorrhagic manifestations did not present significant changes in the levels of VEGF or its soluble receptors [24]. Three studies found no differences in sVEGFR2 levels between severe and nonsevere dengue [17,40,44].

### 2.2. Granulocyte–Macrophage Colony-Stimulating Factor (GM-CSF)

A total of 15 studies analyzed GM-CSF, and 12 found increased levels in dengue patients, half of whom had severe dengue, in contrast with nonsevere dengue and/or healthy controls [12,37,38,45,46,47,48,49,50,51,52,53], including 1 study in children only [48]. The increase in GM-CSF levels is evident mainly in the critical phase [12,37,38,46,52], and elevated GM-CSF levels are associated with clinical manifestations of severe dengue, such as cavitary effusion [46], hypotension [45], and encephalitis [48]. Furthermore, the expression of GM-CSF varied between the primary and secondary infections for each serotype; the upregulation of GM-CSF was found in primary DENV-1 and secondary DENV-2 infections [47]. However, three studies found no significant difference between secondary and primary infection, with or without plasma leakage, or during co-infection with malaria [29,54,55].

### 2.3. Granulocyte Colony-Stimulating Factor (G-CSF)

Eleven studies analyzed G-CSF and seven found increased levels in dengue patients, most of them compared with healthy controls [12,19,31,38,50,51,56]. Three studies found significantly increased levels of G-CSF in severe dengue compared with nonsevere dengue [31,51,56], and one of them demonstrated significantly elevated levels of G-CSF in patients with severe bleeding and deceased patients [31]. However, two studies reported lower levels of G-CSF in dengue patients than in healthy controls [34] and malaria patients [57], and two studies found no significant difference between dengue patients with or without plasma leakage or during coinfection with malaria [29,55].

### 2.4. Transforming Growth Factor Beta (TGF-β)

A total of 13 studies analyzed TGF-β, and 9 found increased levels in dengue patients; most of them compared severe dengue with nonsevere dengue and/or healthy controls [18,57,58,59,63,64,66,67,68], including one study in children only [64]. The range of mean concentrations was 12.6–15,107 pg/mL using the ELISA method [57,58,59,63,66,67].

In addition, one study found decreased intraplatelet levels of TGF-β in dengue with warning signs (DwWS) compared with dengue without warning signs (DwoWS) and healthy controls [35]. TGF-β values peak at the critical phase and remain elevated even during the convalescence phase [58,59,66,67]. Furthermore, increased levels of TGF-β are associated with thrombocytopenia and hypoalbuminemia in patients with plasma leakage [18].

One study showed that TGF-β levels were significantly lower in children with severe dengue than in healthy controls on admission, but they increased on the following days [61]. No significant difference was found in one study comparing severe dengue with and without shock [65] and another study comparing the leakage and no leakage groups [29].

### 2.5. Fibroblast Growth Factor (FGF)

A total of nine studies analyzed FGF, and four studies demonstrated increased FGF levels in dengue patients [19,51,56,62], including one study in children only [62]. FGF values peaked at the febrile phase and remained elevated even in the critical and convalescent phases, mainly in nonsevere dengue [19].

However, four other studies reported the downregulation of FGF in dengue patients compared with healthy controls [29,34,38,60], and in one study, it was associated with elevated aspartate aminotransferase levels [38]. In addition, one study found no difference between nonsevere dengue patients and healthy controls [12].

### 2.6. Platelet-Derived Growth Factor (PDGF)

A total of seven studies analyzed PDGF, and five found decreased levels in dengue patients compared with healthy controls [18,34,35,38,60], including DwWS [35,38] and severe dengue [34]. PDGF levels were lower in the critical phase [18,34,38,60] and secondary infection [38] and were associated with thrombocytopenia [18].

One study reported very high levels of PDGF in dengue patients in the febrile phase in contrast with healthy controls, although significantly lower levels were found in severe dengue patients than in nonsevere dengue patients [19]. In addition, one study found no difference between nonsevere dengue patients and healthy controls [12].

### 2.7. Epidermal Growth Factor (EGF)

A total of six studies analyzed EGF, and five found decreased levels in dengue patients; most of them were severe dengue compared with nonsevere dengue and/or healthy controls [29,31,36,37,62], including two studies in children only [36,62]. EGF levels were lower during the critical phase [36,37].

One study reported significantly elevated EGF levels in dengue patients compared with healthy controls, but decreased EGF levels were associated with hypoalbuminemia and thrombocytopenia during the critical phase [18].

### 2.8. Hepatocyte Growth Factor (HGF)

Six studies analyzed HGF, and four found increased levels in dengue patients, most of them severe dengue compared with nonsevere dengue and/or healthy controls [18,31,33,69], including one study in children only [69].

In addition, one study reported elevated levels of intraplatelet HGF in DwWS [35]. HGF values peaked during the febrile and critical phases [18,33,69], and upregulation was associated with clinical signs of plasma leakage and secondary infection [18]. HGF was identified as a biomarker of severity with an AUC of 0.75 based on ROC curve analysis in the febrile phase [69]. One study found no significant difference between nonsevere dengue patients and healthy controls [12].

## 3. Discussion

We performed an evidence synthesis on the role of GFs in the pathogenesis of dengue based on studies conducted mainly in the last 15 years. Figure 2 summarizes the main roles played by GFs during dengue virus infection.

GFs are proteins that promote or inhibit mitosis and differentiation by binding to receptor tyrosine kinases (RTK), except for TGF-β, which binds to serine/threonine receptors [5]. To perform their functions, GFs can activate multiple receptors with structural similarities, and growth factor receptors (GFR) can have multiple ligands with structural and functional similarities, the only exception being HGF and its receptor Met, which are exclusive to each other [70]. However, there is increasing evidence that viruses can interact with RTKs, modulating their activity to increase cell entry and promote replication into the host [71].

VEGF is the most studied growth factor for dengue and has been suggested as a biomarker of severity [20,25]. This trend of increased levels of severe dengue was maintained regardless of disease classification, age group, or sample type.

The VEGF family includes several members: VEGF-A, VEGF-B, VEGF-C, VEGF-D, VEGF-E, placental growth factor (PlGF), and endocrine-gland derived vascular endothelial growth factor (EG-VEGF). VEGF, secreted by endothelial cells, macrophages, platelets, keratinocytes, leukocytes, dendritic cells, and bronchial and alveolar epithelial cells, plays an important role in vasculogenesis and neoangiogenesis [72]. In vitro studies using human cell lineages KU812, and HMC-1 have indicated that mastocytes are one of the sources of VEGF in dengue. High levels of VEGF have been reported in KU812 and HMC-1 supernatant when infected with DENV-2 in the presence of human dengue-immune serum, and it was even higher in the presence of IL-9 [16]. The biological effects of VEGF are mediated by three RTKs that are expressed in areas of the body where fluid balance is critically regulated and may be localized, such as the choroid plexus of the brain, or systemically, such as resistance vessels [73]. In brief, VEGFR-1 is mainly expressed in monocytes, macrophages, vascular smooth muscle cells, and neuronal cells; VEGFR-2 in endothelial cells; and VEGFR-3 in the lymphatic endothelium [71].

Inflammatory cytokines IL-1β and TNF-α are commonly elevated in dengue cases. These cytokines can increase VEGF production using the NF-κB pathway [30]. VEGF may contribute to inflammation by modulating the expression of P and E selectins as well as integrin-binding adhesion molecules (ICAM-1 and VCAM-1) involved in leukocyte recruitment and causing changes in vascular permeability [73]. Furthermore, coagulation and fibrinolysis mechanisms are activated during DENV infection [74]. In this process, thrombin formation can increase VEGF expression in endothelial cells and enhance its effects through the upregulation of VEGF receptor expression [75]. In addition, VEGF can increase tissue factor expression [76], and the activation of tissue factor pathway is another important component of DENV-related coagulation disorders [77]. However, although one study reported an association between elevated levels of VEGF and D-dimer [30], another study showed that plasma levels of VEGF and its soluble receptors were not associated with the occurrence of bleeding in patients without plasma leakage, suggesting that bleeding is a complication independent of changes in endothelial barrier function [24], as established by the WHO dengue case classification 2009 [11]. In addition, it has been established that VEGF may also be an important element in the pathogenesis of viral diseases, and many viruses seek the positive regulation of VEGF by different means. Some viruses have a homologous structure, and others such as DENV activate inflammatory mediators that upregulate VEGF expression [78].

Several studies have also analyzed soluble VEGF receptors. VEGFR2 is involved in all aspects of normal and pathological vascular endothelial cell biology, and VEGFR1 may be downregulated through sVEGFR1, which binds VEGF and prevents its binding to VEGFR2 [79]. DENV has been shown to downregulate sVEGFR2 production directly by interacting with endothelial cells and producing increased free VEGF that can also be produced by specific T cells activated by DENV [27]. Thus, increased levels of VEGF and expression of surface VEGFR2 result in increased vascular permeability and clinical plasma leakage in severe dengue, as demonstrated in many studies.

Dengue studies included the analysis of two members of the colony-stimulating factor (CSF) superfamily, GM-CSF and G-CSF, and increased levels of both glycoproteins were reported in severe dengue. GM-CSF and G-CSF are produced by a variety of hematopoietic and non-hematopoietic cells (fibroblasts, endothelial, and epithelial cells), and under homeostatic conditions, they are produced by stromal cells within the bone marrow and are detected at very low levels in the serum [80,81].

GM-CSF is essential for responding to the increased demand for granulocytes and macrophages during infection, and its effects are dose-dependent on resident macrophages and/or migrated monocytes and granulocytes. High doses are required to promote the survival, differentiation, and proliferation of monocytes/macrophages, whereas lower doses are required for granulocytes [82]. In the case of dengue, the local elevation of GM-CSF triggers inflammatory macrophages, where DENV replicates efficiently and activates the NLPR3 inflammasome through CLEC5A to induce the massive production of IL-1β and IL-18, which are proinflammatory cytokines associated with severe dengue [83]. Macrophages stimulated by GM-CSF are more susceptible to DENV infection, release more inflammatory cytokines when infected, and their supernatant is a potent inducer of vascular permeability in HMEC-1 cell culture [84]. In addition, another study found that GM-CSF and IFN-γ show antagonistic expression: high levels of GM-CSF antagonized the signals induced by IFN-γ in secondary DENV infections [47].

G-CSF has a broad effect on the innate immune response since it is also a key regulator of the production, differentiation, and release of neutrophil precursors, as well as the modulation of the function of mature neutrophils [81]. Neutrophil extracellular traps (NETs) have been shown to be a defense strategy against DENV infection and are associated with progression to severe dengue [85].

TGF-β is another GF frequently studied in dengue. TGF-β isoforms (TGF-β1, -β2, and -β3) are found only in vertebrates and belong to a family of signaling proteins known as the TGF-β superfamily [86]. TGF-β is required for the proper regulation of cell proliferation, migration, and differentiation during embryonic development and for maintaining tissue homeostasis in adults [87]. Once activated, TGF-β has pleiotropic effects that regulate effector and regulatory CD4+ T cell responses depending on the cytokine environment, exerting an inflammatory and suppressive role [88]. It has been established that TGF-β can inhibit differentiation of T-helper type 1 (Th1) and type 2 (Th2) cells while upregulating cell lineage specification from naive CD4+ T cells to Th17 or regulatory T (Treg) cells [89]. Although Treg cells are elevated in dengue cases, no correlation has been established between them and TGF-β levels or disease severity [63], whereas another study found an association between Th17 cells and dengue severity in secondary infections [90]. In addition, TGF-β can induce the expression of soluble adhesion molecules such as ICAM-1, VCAM-1, and E-selectin in endothelial cells [91].

It has been determined that the highest concentrations of TGF-β are found in the alpha granules of platelets, which are the main regulator of the concentration of TGF-β in blood plasma. However, the mechanisms that regulate this secretion are still unclear [92]. One study showed a decrease in the intraplatelet concentration in DwWS [35], and most studies reported elevated serum TGF-β levels in severe disease, including an association with plasma leakage [18]. In vitro studies have suggested that TFG-β plays an important role in DENV replication, indicating it as a potential therapeutic target. Macrophages infected with DENV-2 and treated with TFG-β receptor inhibitor had an intense reduction in viral load after 48h of incubation. This effect was dose dependent and it was reversed when the cells were treated with recombinant TFG-β [93]. In addition, a study demonstrated that a combination of TGF-β1 and cytotoxic T-lymphocyte antigen 4 (CTLA-4) was associated with susceptibility to severe dengue [94].

Studies have demonstrated decreased levels of platelet-derived growth factor (PDGF) and epidermal growth factor (EGF) in severe dengue compared with nonsevere dengue, which are associated with thrombocytopenia. Both GFs are involved in tissue repair, and their receptors are expressed in cells that participate in this process, such as endothelial cells, fibroblasts, smooth muscle cells, keratinocytes, mesangial cells, and mesenchymal stem cells [95,96]. Platelets are known to release several GFs (PDGF, EGF, TGF-β, FGF, etc.) during the repair of vascular injury or stimulation of inflammation, which could explain the decreased levels of PDGF and EGF in thrombocytopenia caused by dengue [97]. Alternatively, it is believed that the reduced levels of PDGF and EGF in dengue patients could reflect their consumption in the endothelial repair process [36].

HGF is another growth factor involved in dengue pathogenesis. The HGF-Met pathway plays a fundamental role in the regulation of liver growth and regeneration, and it also supports the protection and regeneration of kidney, lung, nervous system, cardiovascular, skin, and gastrointestinal tissues [98]. HGF contributes to tissue repair by promoting the resolution of inflammation, and is produced by platelets, monocytes, neutrophils, and mast cells. Neutrophils contain preformed HGF in secretory granules that are released upon cell stimulation, unlike other immune cells that express HGF only after activation [99]. HGF induces monocyte/macrophage activation and directional migration, regulates dendritic cell function, decreases IL-6 production, increases IL-10 production, and promotes the maturation of Tregs [100,101,102,103,104].

HGF is expressed in nonparenchymal liver cells, such as Kupffer cells, endothelial cells, and Ito cells. Serum HGF levels increase in response to liver damage and utilize the autocrine, paracrine, and endocrine modes of HGF-MET signaling [105]. Increased levels of HGF are found in liver diseases, such as acute hepatitis, chronic hepatitis, liver cirrhosis, hepatocellular carcinoma, and fulminant hepatic failure [105]. Liver involvement is common in DENV infection [106], and it has been suggested that increased HGF expression occurs in response to DENV-induced liver injury [69]. In relation to the involvement of other organs due to arbovirus infection, one study found elevated serum HGF levels in the acute phase in patients with neurological complications [107], and another study reported decreased serum HGF levels in infants with Zika virus-related congenital neurological disease [108].

Finally, although some studies have demonstrated the upregulation of FGF in dengue patients, other studies have reported reduced levels. The FGF family includes 18 secreted signaling proteins consisting of autocrine/paracrine FGFs (also called canonical FGFs) and endocrine FGFs that activate 4 receptor tyrosine kinases (FGFR) 1–4. These are expressed in almost all tissues; play a fundamental role in early embryonic differentiation and development; and function as homeostatic factors for tissue maintenance, repair, regeneration, and metabolism [109]. Canonical FGFs include five subfamilies that use heparan sulfate as a co-factor for binding to FGFR, and the FGF1 subfamily is composed of FGF1 (also known as acidic FGF) and FGF2 (also known as basic FGF). Elevated levels of FGF2 have been implicated in the pathogenesis of several diseases characterized by a dysregulated angiogenic/inflammatory response because FGF2-stimulated endothelial cells upregulate adhesion molecules, proinflammatory cytokines/chemokines, and GFs (such as VEGF). However, prolonged exposure to FGF2 can inhibit monocyte/macrophage adhesion, transendothelial migration of CD4+ T cells, and tissue factor expression in endothelial cells [110]. A mechanism by which DENV uses the FGFR4 signaling pathway to promote its spread in the host was recently described: At an early stage when FGFR levels are high, DENV RNA replication is favored; at a late stage when replication is at a steady state and FGFR levels are low, virion production is favored by enhancing the maturation of DENV virions through prM cleavage [111].

Dengue does not have a specific treatment, but several preclinical and clinical studies are being conducted using two approaches. The largest number of studies focus on antivirals directed against viral proteins that would have less toxicity but a high possibility of developing resistance [112]. The other approach is aimed at inhibiting the host components used by DENV for its dissemination, which would have a greater risk of toxicity but the advantage of a higher genetic barrier to resistance [113].

This review attempts to show how the growth factor pathway performs its function of restoring tissue homeostasis during DENV infection. However, as mentioned above, there is evidence that this virus can use this pathway to facilitate its entry into cells and promote viral replication. Thus, the use of drugs to target GFs and their tyrosine kinase activities is an alternative. A range of these drugs is already used in oncology, and their repurpose for neglected diseases would be advantageous as it would reduce the time needed to develop therapies for them de novo [113]. In recent years, several studies have been published on anti-GFR tyrosine kinase inhibitors (TKIs) in dengue, such as dasatinib [114], sunitinib/erlotinib [115], gefitinib [116], brivanib [117], afatinib-derivative (L3) [118], sunitinib/anti-TNF antibody [119], and TGF-β receptor 1 and 2 inhibitors (GW788388) [93].

We report the available evidence on GFs in dengue patients. However, some limitations of this review should be considered. There was heterogeneity in the studies in relation to the clinical classification of patients, use of days of illness or days in relation to admission to group patients, type of sample, time of sample collection, and technique for measuring GFs that could have led to inconsistent associations. In addition, the estimation of effect sizes and the use of healthy controls to compare them with groups of patients are aspects that should always be considered in future studies. The lack of reporting of the results of GFs in several studies could also have led to incorrect conclusions.

## 4. Materials and Methods

This scoping review was conducted in accordance with the Joanna Briggs Institute’s methodology [120]. The protocol for this study was registered in the Open Science Framework [121]. The report followed the Preferred Reporting Items for Systematic Reviews and Meta-analyses extension for Scoping Review (PRISMA-ScR) [9].

### 4.1. Search Strategies

A systematic literature search of MEDLINE and Embase databases was performed using keywords and index terms related to “dengue” and “growth factors (VEGF, GM-CSF, G-CSF, TGF, FGF, PDGF, EGF, HGF, IGF)” and Boolean operators (“AND” and “OR”). The detailed search strategies are presented in Appendix A.

### 4.2. Elegibilty Criteria

This review considered all peer-reviewed published studies, with no limits on publication dates. The following study types were considered eligible: cross-sectional, prospective, and retrospective. Conference abstracts, case reports, case series, and reviews were considered to be ineligible. Studies published in English, Spanish, or Portuguese were included.

### 4.3. Study Selection

Identified citations were uploaded to the EndNote X9 software (Clarivate Analytics, Philadelphia, PA, USA), and duplicates were removed. Publications were screened by title and abstract for possible inclusion by two double-blind reviewers (VEFS and RCdL). Disagreements between reviewers were resolved through discussion or by a third reviewer (ELdA). The full texts were then searched for articles that met the inclusion criteria. The results of the study selection are presented in a flow diagram according to PRISMA-ScR.

### 4.4. Data Extraction

The following information was extracted from eligible studies: (1) general information (first author, journal, year, location, title, objective, study design); (2) study population (sample size, control group, age group, classification of dengue patients, type of infection, serotype); and (3) growth factor (type of sample, laboratory technique, results), using a custom data extraction tool.

## 5. Conclusions

This scoping review summarizes the evidence on the fundamental role of the GF pathway in restoring tissue homeostasis during DENV infection; repairing vascular injury and other damaged tissues; and modulating the expression and differentiation of monocytes, macrophages, lymphocytes, and neutrophils that could favor an intense inflammatory response and cause increased vascular permeability. Thus, increased levels of VEGF, GM-CSF, G-CSF, TGF-β, and HGF, as well as decreased levels of PDGF and EGF, were observed in severe dengue in most studies. VEGF and HGF have been identified as biomarkers of severity. In addition, there is evidence that dengue virus can use the GFs pathway to facilitate its entry into the cell and promote viral replication, and the use of inhibitors of GFs and their tyrosine kinase activities are an alternative treatment for dengue. Research efforts are still required for a better understanding of the role of the GF pathway, which will allow the design of studies for new treatment protocols.

## Figures and Tables

**Figure 1 pathogens-11-01179-f001:**
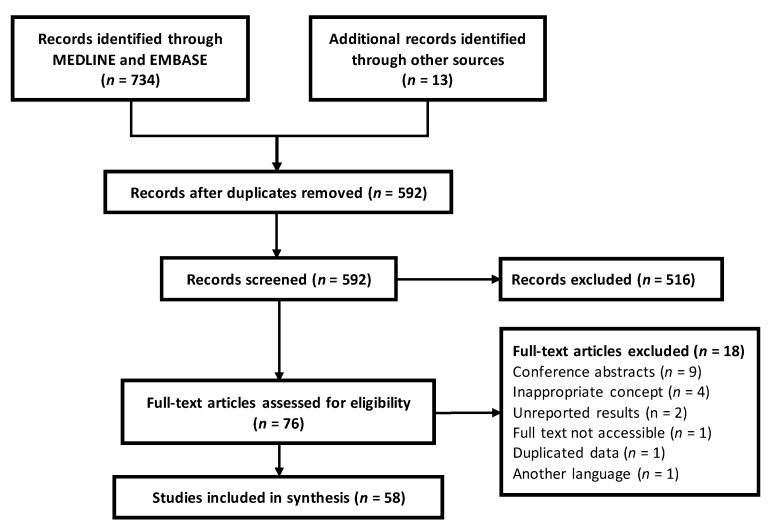
PRISMA–ScR flow diagram of bibliographic search and screening articles.

**Figure 2 pathogens-11-01179-f002:**
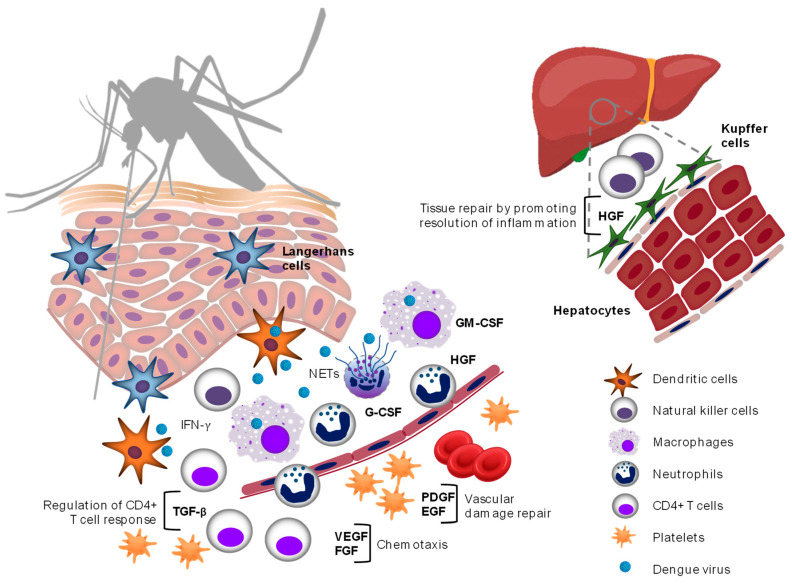
Role of GFs in pathogenesis of dengue. This scheme represents the main roles that GFs play during dengue virus infection that could favor an intense inflammatory response and increased vascular permeability in severe dengue. Platelets are one of the main sources of VEGF, TGF-β, PDGF, EGF, and HGF. Additionally, neutrophils contain pre-formed HGF in secretory granules that is released upon cell stimulation. VEGF and FGF contribute to inflammation by upregulating leukocyte and endothelial cell adhesion molecules. TGF-β regulates effector and regulatory CD4+ T cell responses. GM-CSF and G-CSF are produced by hematopoietic and non-hematopoietic cells to respond to monocyte/macrophage and neutrophil demand, respectively. Elevated levels of PDGF and EGF are required for the repair of vascular injury. HGF contributes to liver tissue repair by promoting the resolution of inflammation.

**Table 1 pathogens-11-01179-t001:** Summary of the characteristics and results of the included studies.

First Author	Location	Study Design	Age Group	Dengue Patients	Measurement of GFs *		Sample	Laboratory Technique
Becquart et al., 2010 [12]	GA	Cross-sectional	Adults	DF: 36	VEGF: GM-CSF: G-CSF: PDGF: HGF: FGF:	↑ DF compared with HC (a, b, c) ↑ DF compared with HC (a, b, c) ↑ DF compared with HC (a, b, c) no difference between DF and HC no difference between DF and HC no difference between DF and HC	Plasma	Luminex
Conroy et al., 2014 [13]	CO	Case-control	Adults and children	Dengue: 113	VEGF:	↑ dengue patients compared with HC (a)	Serum	ELISA
Conroy et al., 2015 [14]	CO	Case-control	Adults and children	DF: 65 DHF: 46	VEGF:	no difference between DF and DHF (a)	Serum	ELISA
Del Moral-Hernández et al., 2014 [15]	MX	Cross-sectional	Adults	DF: 70 DHF: 80	VEGF:	↑ DHF and DF compared to HC	Serum	ELISA
Furuta et al., 2012 [16]	VN	Cohort	Children	DF: 19 DHF: 84	VEGF:	↑ DHF compared with DF and HC	Plasma	ELISA
Garcia et al., 2019 [17]	CO	Cross-sectional	Children	DwWS: 15 SD: 16	VEGF:	↑ SD and DwWS compared with HC	Plasma	ELISA
Her et al., 2017 [18]	SG	Cohort	Adults	Non-leakage: 30 Mild leakage: 30 Significative leakage: 30	VEGF: TGF-β: PDGF: EGF: HGF:	↑ dengue patients compared with HC (a, b, c) ↑ TGF-β was associated with plasma leakage ↓ PDGF was associated with thrombocytopenia ↑ dengue patients compared with HC (a, b, c) ↑ dengue patients with significative leakage (b)	Plasma	Luminex
Kumar et al., 2012 [19]	SG	Cohort	Adults	DF: 44 DHF: 18	VEGF: G-CSF: PDGF: FGF:	↑ dengue patients compared with HC, DF > DHF (a) ↑ dengue patients compared with HC, DF > DHF (a) ↑ dengue patients compared with HC, DF > DHF (a) ↑ dengue patients compared with HC (a, b, c)	Serum	Luminex
Low et al., 2018 [20]	MY	Cohort	Adults	Nonsevere: 53 Severe: 29	VEGF:	↑ severe compared with nonsevere dengue (a)	Plasma	ELISA
Mangione et al., 2014 [21]	VN	Case-control	Children	DF: 6, DHF: 18, DSS: 33	VEGF:	↑ DSS compared with OFI no difference between DF, DHF and HC	Plasma	Luminex
Mutiara et al., 2019 [22]	ID	Cross-sectional	Adults and children	DHF: 50	VEGF:	↑ DHF compared with HC (a, b, c)	Plasma	ELISA
Nanda et al., 2021 [23]	TW	Cross-sectional	Adults	Nonsevere: 10 Severe: 20	VEGF:	↑ severe dengue with comorbidity compared to nonsevere and severe dengue without comorbidity	Plasma	Luminex
Orsi et al., 2014 [24]	BR	Cross-sectional	Adults	DF: 59 DHF: 10	VEGF:	↑ DHF compared with DF	Plasma	ELISA
Patra et al., 2019 [25]	IN	Cross-sectional	Adults	DwoWS: 54, DwWS: 25, SD: 19	VEGF:	↑ SD and DwWS compared with DwoWS and HC (b)	Serum	ELISA
Singla et al., 2016 [26]	IN	Cohort	Children	DwoWS: 21 DwWS: 30, SD: 46	VEGF:	↑ SD compared with DwWS (b)	Plasma	Luminex
Srikiatkhachorn et al., 2007 [27]	TH	Cohort	Children	DF: 22 DHF: 23	VEGF:	↑ DHF compared with DF (b)	Plasma	ELISA
Thakur et al., 2016 [28]	IN	Cross-sectional	Adults	Nonsevere: 58 Severe: 48	VEGF:	↑ severe compared with nonsevere dengue and HC (b, c)	Serum	ELISA
Tramontini et al., 2017 [29]	BR	Cross-sectional	Adults and children	Non-leakage: 29 Leakage: 28	VEGF: GM-CSF: G-CSF: EGF: FGF:	↑ leakage group compared with HC no difference between leakage, non-leakage and HC no difference between leakage, non-leakage and HC ↓ leakage group compared to non-leakage and HC ↓ dengue patients compared with HC	Serum	Luminex
Tseng et al., 2005 [30]	TW	Cross-sectional	Adults	DF: 39 DHF: 14	VEGF:	↑ DHF compared to DF and HC	Plasma	ELISA
van de Weg et al., 2013 [31]	BR	Cross-sectional	Adults and children	DwoWS: 50 DwWS: 49, SD: 33	VEGF: G-CSF: EGF: HGF:	↑ SD compared with DwWS and DwoWS ↑ SD compared with DwWS and DwoWS ↓ SD compared with DwWS and DwoWS ↑ SD compared with DwWS and DwoWS	Serum	Luminex
van de Weg et al., 2014 [32]	BR	Cross-sectional	Adults and children	Non-leakage: 56 Leakage: 49	VEGF:	↑ leakage and non-leakage groups compared with HC	Serum	ELISA
Yong et al., 2017 [33]	MY	Cohort	Adults	DwoWS: 43 DwWS: 92, SD: 6	VEGF: HGF:	↑ SD compared with DwWS and DwoWS (a, b) ↑ SD and DwWS compared with DwoWS (a)	Plasma	Luminex
Appanna et al., 2012 [34]	MY	Cohort	Adults	DF: 13 DHF: 29	VEGF: G-CSF: PDGF: FGF:	↓ DHF compared with HC (b) ↓ DHF and DF compared with HC (b, c) ↓ DHF compared with HC (b) ↓ DHF compared with HC (b)	Serum	Luminex
Barros et al., 2019 [35]	BR	Cross-sectional	Adults	DwoWS:53 DwWS:31	VEGF: TGF-β: PDGF: HGF:	↓ DwWS compared with DwoWS and HC ↓ DwWS compared with DwoWS and HC (intraplatelet) ↓ DwWS compared with DwoWS and HC ↑ DwWS compared with DwoWS and HC (intraplatelet)	Serum	Luminex
Butthep et al., 2012 [36]	TH	Cohort	Children	DF: 51 DHF: 196	VEGF: EGF:	↓ DHF compared with DF and OFI (a, b) ↓ DHF compared with DF and OFI (b)	Not specified	Chemiluminescent immunoassay
Chagan-Yasutan et al., 2013 [37]	PH	Cohort	Adults	Dengue: 65	VEGF: GM-CSF: EGF:	↓ dengue patients compared with HC (b, c) ↑ dengue patients compared with HC (b) ↓ dengue patients compared with HC (b, c)	Plasma	Luminex
Rathakrishnan et al., 2012 [38]	MY	Cohort	Adults	DwoWS: 11 DwWS: 29, SD: 4	VEGF: GM-CSF: G-CSF: PDGF: FGF:	↓ DwWS compared with DwoWS and HC (a, b) ↑ DwWS compared with HC (b) ↓ DwoWS and DwWS compared with HC (a, b, c) ↓ DwWS compared with DwoWS and HC (a, b, c) ↓ DwoWS and DwWS compared with HC (a, b, c)	Serum	Luminex
Sathupan et al., 2007 [39]	TH	Cohort	Children	DF: 15 DHF: 26	VEGF:	↓ DHF compared with DF and HC (b)	Serum & plasma	ELISA
Seet et al., 2009 [40]	SG	Case-control	Adults	DF: 34 DHF: 26	VEGF:	↓ DHF and DF compared with HC and OFI	Serum	ELISA
Figueroa et al., 2016 [41]	CO	Case-control	Adults and children	DF: 30 DHF: 30	VEGF:	no difference between DF and DHF (a)	Serum	ELISA
Misra et al., 2014 [42]	IN	Cross-sectional	Adults	DF: 12 DHF: 9	VEGF:	no difference between DHF and DF	Serum	ELISA
Yacoub et al., 2016 [43]	VN	Cohort	Adults and children	Non-leakage: 33 Leakage: 37	VEGF:	no difference between leakage and non-leakage group	Plasma	Luminex
Kalita et al., 2015 [44]	IN	Cross-sectional	Adults	DF: 20 DHF: 7	VEGF:	no difference between dengue patients and HC	Serum	ELISA
Bozza et al., 2008 [45]	BR	Cross-sectional	Adults	Mild: 20 Severe: 39	GM-CSF:	↑ severe compared to mild dengue	Plasma	Luminex
Gowri et al., 2021 [46]	IN	Cross-sectional	Adults and children	DF: 21, DHF: 35, DSS: 20	GM-CSF:	↑ dengue patients compared with HC (b) ↑ secondary compared to primary infection	Plasma	ELISA
Gowri S et al., 2021 [47]	IN	Cross-sectional	Adults and children	DF: 21, DHF: 35, DSS: 20	GM-CSF:	↑ DSS compared with DHF and DF (DENV-2)	Plasma	ELISA
Li et al., 2017 [48]	CN	Cross-sectional	Children	Dengue encephalitis: 29	GM-CSF:	↑ dengue encephalitis compared with HC	Serum & CSF	Luminex
Maneekan et al., 2013 [49]	TH	Cross-sectional	Not reported	Dengue: 20	GM-CSF:	↑ dengue patients compared to HC	Serum	Luminex
Oliveira et al., 2017 [50]	BR	Cohort	Adults and children	DF: 25 DHF: 19	GM-CSF: G-CSF:	↑ dengue patients compared with HC ↑ dengue patients compared with HC	Serum	Cytokine antibody array
Patro et al., 2019 [51]	IN	Cross-sectional	Adults	Nonsevere: 73 Severe: 12	GM-CSF: G-CSF: FGF:	↑ severe compared with nonsevere dengue and HC ↑ severe compared with nonsevere dengue ↑ severe compared with nonsevere dengue	Plasma	Luminex
Puc et al., 2021 [52]	TW	Cross-sectional	Adults	DwoWS: 128, DwWS: 103, SD: 53	GM-CSF:	↑ SD compared with DwWS, DwoWS and HC (b, c)	Plasma	Cytometric bead array
Wang et al., 2019 [53]	TW	Cross-sectional	Adults	DF: 40 DHF: 20	GM-CSF:	↑ DHF compared with DF and HC	Plasma	Luminex
Meena et al., 2020 [54]	IN	Cross-sectional	Adults and children	not specified	GM-CSF:	no difference between dengue patients and HC	Serum	Luminex
Mendonça et al., 2015 [55]	BR	Cross-sectional	Adults	Dengue: 30	GM-CSF: G-CSF:	no difference between dengue, malaria and coinfection no difference between dengue, malaria and coinfection	Plasma	Luminex
Cui et al., 2016 [56]	SG	Cross-sectional	Adults	DF: 81 DHF: 81	G-CSF: FGF:	↑ DHF compared with DF ↑ DHF compared with DF	Serum	Luminex
Halsey et al., 2016 [57]	PE	Cross-sectional	Adults and children	Dengue: 51	G-CSF: TGF-β:	↓ dengue and coinfection compared to malaria ↑ dengue compared with malaria or coinfection	Serum	Luminex
Agarwal et al., 1999 [58]	IN	Cross-sectional	Adults and children	DF: 18 DHF: 61	TGF-β	↑ dengue patients compared to HC ↑ DHF compared with DF (b)	Serum	ELISA
Azeredo et al., 2006 [59]	BR	Cross-sectional	Adults	Mild: 31 Severe: 19	TGF-β	↑ severe compared with mild dengue and HC (a, b, c)	Plasma	ELISA
Pandey et al., 2015 [60]	IN	Cross-sectional	Adults and children	DwoWS: 37, DwWS: 53, SD: 121	TGF-β:	↑ dengue patients compared with HC ↑ SD and DwWS compared with DwoWS (b)	Plasma	ELISA
Tillu et al., 2016 [61]	IN	Cross-sectional	Adults and children	DwoWS: 31 DwWS: 20	TGF-β	↑ DwWS compared with DwoWS and HC	Plasma	Luminex
Patra et al., 2019 [62]	IN	Cross-sectional	Not reported	DwoWS: 32, DwWS: 21, SD: 9	TGF-β:	↑ SD and DwWS compared with DwoWS and HC (b)	Serum	ELISA
Jayaratne et al., 2018 [63]	LK	Cohort	Adults	DF: 27 DHF: 26	TGF-β:	↑ DHF compared to DF but decrease in both compared with HC	Plasma	ELISA
Laur et al., 1998 [64]	PF	Cohort	Children	DF: 33 DHF: 17	TGF-β:	↑ DHF compared with DF and HC (a)	Plasma	ELISA
Djamiatun et al., 2011 [65]	ID	Cohort	Children	DHF: 71	TGF-β:	↓ DHF compared with HC (b)	Plasma	ELISA
Malavige et al., 2012 [66]	LK	Cross-sectional	Adults	DHF: 112	TGF-β	no difference between DHF with shock and without shock	Plasma	Luminex
Harenberg et al., 2016 [67]	Multi-center	Clinical trial	Children	Nonsevere: 154 Severe: 52	EGF: FGF:	↓ severe dengue compared with nonsevere dengue ↑ dengue compared to baseline	Serum	Luminex
Dhenni et al., 2021 [68]	ID	Cross-sectional	Adults and children	DF: 43	PDGF: FGF:	↓ DF compared with chikungunya and HC ↓ DF compared to chikungunya and HC	Plasma	Luminex
Voraphani et al., 2010 [69]	TH	Cohort	Children	DF: 17 DHF: 10	HGF:	↑ DHF and DF compared with HC (a, b)	Serum	ELISA

BR, Brazil; CN, China; CO, Colombia; GA, Gabon; ID, Indonesia; IN, India; LK, Sri Lanka; MX, Mexico; MY, Malaysia; PE, Peru; PF, French Polynesia; PH, Philippines; SG, Singapore; TH, Thailand; TW, Taiwan; VN, Viet Nam. VEGF, vascular endothelial growth factor; GM-CSF, granulocyte–macrophage colony-stimulating factor; G-CSF, granulocyte colony-stimulating factor; TGF-β, transforming growth factor beta; PDGF, platelet-derived growth factor; EGF, epidermal growth factor; FGF, fibroblast growth factor. WHO dengue case classification 1997: DF, dengue fever; DHF, dengue hemorrhagic fever; DSS, dengue shock syndrome. WHO dengue case classification 2009: DwoWS, dengue without warning signs; DwWS, dengue with warning signs; SD, severe dengue. HC, healthy controls; OFI, other febrile infections. (*) The phase where significant change was found is indicated when this information is reported: (a) febrile, (b) critical, (c) convalescent.

## Data Availability

Not applicable.

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
