# Peer review of "The Role of Growth Factors in the Pathogenesis of Dengue: A Scoping Review"

_pathogens, 2022, doi:10.3390/pathogens11101179_

Round 1

Reviewer 1 Report

This review systematically summarizes the role of growth factors in the pathogenesis of dengue, and provides some hints for the treatment of dengue. The article is novel, but it cannot meet the requirements for publication, and there are still the following problems:

1. Language problems

Many sentences in this article are not smooth and verbose, and there are grammatical problems such as incorrect subject-verb collocation. Such as Line 8, "Growth factors (GFs) have a role in tissue repair and modulation of the expression of inflammatory cells in damage caused by pathogens"; Line 10 "considered" should be "considers"; Line 95, "most of them severe" dengue compared to non-severe dengue and/or healthy controls"; Line 117 "without plasma leakage with or without hemorrhagic manifestations"; Line 215 "bind"; Line 414 "are"; Line 415 "role" should be "the role"; Line 416 "allow" should be "allows", etc. I hope that the author will carefully revise every grammatical problem found and check the full text.

2. Format problems

Problems with abbreviations: Some abbreviations do not need to appear, such as Line 16 "TGF-β and PDGF" does not appear again in the abstract and can be omitted; some full names and abbreviations appear multiple times, such as Line 97 and Line 195 "AUC"; some abbreviations appear without full names, such as Line 111 “sVEGFR” and Line 335 “FGFRs”; some abbreviations have appeared before, such as Line 51 and 333 “FGF”.

The problem of unified format: whether to add "," before "and" (such as Line 16, 25, and 71 has ","; Line 104 and 111 does not have ","). In Table 1, the capitalization of the text in each column should be unified, such as "Adults" in "Age group" and "serum" in "Sample". In Figure 1, the spacing between "=" and "592" in the two "(n = 592)" is not uniform. Authors should carefully check the format of the article.

3. Line 363-366: These sentences should be adjusted to Line 360

Author Response

"Please see the attachment 

We appreciate the reviewers' questions and comments beforehand, which will undoubtedly contribute to a better understanding of our manuscript.

Reviewer 1

This review systematically summarizes the role of growth factors in the pathogenesis of dengue and provides some hints for the treatment of dengue. The article is novel, but it cannot meet the requirements for publication, and there are still the following problems:

  1. Language problems

Many sentences in this article are not smooth and verbose, and there are grammatical problems such as incorrect subject-verb collocation. Such as Line 8, "Growth factors (GFs) have a role in tissue repair and modulation of the expression of inflammatory cells in damage caused by pathogens"; Line 10 "considered" should be "considers”; Line 95, "most of them severe" dengue compared to non-severe dengue and/or healthy controls"; Line 117 "without plasma leakage with or without hemorrhagic manifestations"; Line 215 "bind"; Line 414 "are"; Line 415 "role" should be "the role"; Line 416 "allow" should be "allows", etc. I hope that the author will carefully revise every grammatical problem found and check the full text.

Authors' response: We appreciate the reviewers' constructive criticism. We want to inform you that all questions and suggestions regarding your general appreciation of the manuscript have been changed as requested in red text. Very grateful for your comments.

We apologize for the English, which made our manuscript difficult to understand. We want the manuscript to have the best possible writing and we send a company to review the entire manuscript, certified attached. Hope it's now more understandable

  1. Format problems

Problems with abbreviations: Some abbreviations do not need to appear, such as Line 16 "TGF-β and PDGF" does not appear again in the abstract and can be omitted; some full names and abbreviations appear multiple times, such as Line 97 and Line 195 "AUC"; some abbreviations appear without full names, such as Line 111 “sVEGFR” and Line 335 “FGFRs”; some abbreviations have appeared before, such as Line 51 and 333 “FGF”.

Authors' response: Thanks. Corrections were made

The problem of unified format: whether to add "," before "and" (such as Line 16, 25, and 71 has ","; Line 104 and 111 does not have ","). In Table 1, the capitalization of the text in each column should be unified, such as "Adults" in "Age group" and "serum" in "Sample". In Figure 1, the spacing between "=" and "592" in the two "(n = 592)" is not uniform. Authors should carefully check the format of the article.

Authors' response: Thanks. Corrections were made

  1. Line 363-366: These sentences should be adjusted to Line 360

Authors' response: Thanks. Clarification was made

Reviewer 2 Report

In this study, Fiestas et al. reviewed cross-sectional, prospective, and retrospective studies that evaluated different growth factors in subjects with dengue infections. This study found that increased levels of Vascular Endothelial Growth Factor (VEGF), Granulocyte Macrophage Colony Stimulating Factor (GM-CSF), Granulocyte Colony Stimulating Factor (G-CSF), and Transforming Growth Factor beta (TGF-β) were evidenced in severe dengue while VEGF, GM-CSF, and Hepatocyte Growth Factor (HGF) were identified as severity biomarkers.

I like the way of describing each growth factor in the text and the table referring to each study. The authors included only human studies measuring different growth factors. I wonder whether the authors have considered including in the text the principal findings of growth factors from basic studies in dengue. This will be valuable for figure 2 and it may already cover in the discussion but is not highlighted.     

Minor improvements needed:

There are different classifications for dengue. I wonder whether the authors could group the findings by the different classifications (for example WHO 1997 and WHO 2009).   

Do the virus serotypes contribute to the differences observed in the growth factor levels?

In my opinion, it would be useful to know the range concentration of each growth factor. Those may be different depending on the method used to test (ELISA vs Luminex), the sample tested (plasma vs serum), or the age of the patients. I think the authors could discuss it briefly in the text.

Author Response

Can be improved results and conclusions

In this study, Fiestas et al. reviewed cross-sectional, prospective, and retrospective studies that evaluated different growth factors in subjects with dengue infections. This study found that increased levels of Vascular Endothelial Growth Factor (VEGF), Granulocyte Macrophage Colony Stimulating Factor (GM-CSF), Granulocyte Colony Stimulating Factor (G-CSF), and Transforming Growth Factor beta (TGF-β) were evidenced in severe dengue while VEGF, GM-CSF, and Hepatocyte Growth Factor (HGF) were identified as severity biomarkers.

I like the way of describing each growth factor in the text and the table referring to each study. The authors included only human studies measuring different growth factors. I wonder whether the authors have considered including in the text the principal findings of growth factors from basic studies in dengue. This will be valuable for figure 2 and it may already cover in the discussion but is not highlighted.

Authors' response Thank you very much for your comments. We appreciate the reviewers' constructive criticism. We want to inform you that all questions and suggestions regarding your general appreciation of the manuscript have been changed as requested in red text.

In the topic of discussion, we try to include the most relevant studies on the subject, although the studies on growth factors are still scarce. In the new version of the manuscript, the information of some basic studies that had already been included was highlighted and some new ones were added.

In this way, we now highlight the role that some inflammatory cytokines (Line 254) and add information about the role that mast cells could have in the production of VEGF (Line 244).

We also added information from in vitro studies that determined that macrophages stimulated by GM-CSF are more susceptible to DENV infection, release more inflammatory cytokines when infected and are a powerful inducer of vascular permeability (Line 294).

Information from another in vitro study suggesting a role for TGF in DENV replication and the effect of using an TGF-β receptor inhibitor was also added (Line 324).

In addition, the potential effect that growth factors could have on severe dengue was highlighted in the legend of figure 2.

Minor improvements needed:

There are different classifications for dengue. I wonder whether the authors could group the findings by the different classifications (for example WHO 1997 and WHO 2009).  

Authors' response: Very grateful for your comments.

The corresponding analysis was made, and it was found that the trend of increasing VEGF levels in severe dengue was maintained regardless of the clinical classification. This finding was included in the results (Line 102) and discussion (Line 238).

However, for the other growth factors, no inference could be made due to the smaller number of studies.

Do the virus serotypes contribute to the differences observed in the growth factor levels?

Authors' response: In the article it was mentioned that only half of the studies had information on the serotype and in half of them DENV-2 was more frequent. However, for each growth factor, studies with information from another serotype were insufficient to make any inference.

In my opinion, it would be useful to know the range concentration of each growth factor.

Authors' response: Only half of the studies had information on growth factor concentrations which had already been considered within the limitations of the study.

However, the concentration ranges of the studies that had this information were included in the new version of the manuscript, so it was only possible to include this information for VEGF (Line 102) and TGF-β (Line 166).

Those may be different depending on the method used to test (ELISA vs Luminex), the sample tested (plasma vs serum), or the age of the patients. I think the authors could discuss it briefly in the text.

Authors' response: Unfortunately, this analysis could not be adequately performed due to the insufficient number of studies. In the case of VEGF, which had the largest number of studies in this review, a greater variability was found in adults and serum samples analyzed by ELISA method.

However, it is necessary to consider that this variability may be associated with other variables not considered, such as the sensitivity of the different ELISA kits used, sample collection time, etc., for which it would be difficult to establish a precise inference. Therefore, it has been pointed out as a limitation of the study.

Round 2

Reviewer 1 Report

This review systematically summarizes the role of growth factors in the pathogenesis of dengue, and provides some hints for the treatment of dengue. The manuscript has met the requirements for publication.